# Sex and Age Differences in Association between Physical Activity and Metabolic Syndrome: Results from NHANES 2003–2006

**DOI:** 10.3390/healthcare11081059

**Published:** 2023-04-07

**Authors:** Hanying Li, Henry S Lynn, Vadim Zipunnikov

**Affiliations:** 1Department of Biostatistics, School of Public Health, Fudan University, Shanghai 200000, China; 2Department of Biostatistics, Johns Hopkins University, Baltimore, MD 21205, USA

**Keywords:** metabolic syndrome, physical activity, sex, age

## Abstract

Objective: To examine whether and how sex and age modify the association between accelerometer-based physical activity (PA) and metabolic syndrome (MetS) among American (US) adults. Method: Adults aged ≥20 years old who participated in the mobile center examination during 2003–2006 in the National Health and Nutrition Examination Survey were included for analysis. The total minutes per day of moderate-to-vigorous PA (MVPA) was estimated using ActiGraph. Multivariable logistic regression was used to estimate the odds ratio (OR) of having MetS at an increasing MVPA time. The modification effects of gender and age on the association between MetS and MVPA time were examined by testing for two-way and three-way interaction terms of MVPA time, sex, and age in the model after adjusting for relevant covariates. Results: The prevalence of MetS generally decreased with the MVPA time and was lower in females than in males, although the sex difference varied across age groups. After adjusting for demographic and lifestyle covariates, there was a significant sex difference in how an increased MVPA time lowered the odds of MetS. This interactive effect also varied with age. MVPA benefitted young and middle-age populations up until about 65 years old for both sexes, and the protective effect weakened with age. Although the effect of MVPA was stronger for males than females at young ages, the rate at which it attenuated was quicker in males. The OR of MetS between males and females per unit change of MVPA time was 0.73 (95% CI: [0.57, 0.93]) at age = 25 years, compared to OR = 1.00 (95% CI: [0.88, 1.16]) at age = 60 years. Before the age of 50, the gender difference in the protective effect on MetS was larger at low MVPA levels and became smaller at higher MVPA levels. The male advantage was quite stable with an increasing MVPA time for ages 50–60, and no longer significant at older ages. Conclusions: Young and middle-age populations benefitted from MVPA, lowering the risk of MetS for both sexes. A longer MVPA time was associated with a greater decrease in the risk of MetS in young men than in women, but the sex difference reduced with age and was no longer apparent in older populations.

## 1. Introduction

Metabolic syndrome (MetS), a pathologic state of abnormal clustering of various metabolic components, is a general term for clinical symptoms that include central and abdominal obesity, systemic hypertension, insulin resistance (or type 2 diabetes), and atherosclerotic dyslipidemia [1,2,3,4]. MetS is also associated with diabetes, cardiovascular and cerebrovascular diseases, and all-cause mortality [5]. The prevalence of MetS varies worldwide, partly because of the different criteria used in its definition. Nonetheless, the incidence of MetS among adults has increased along with that of obesity and of type 2 diabetes. Among American (US) adults, the prevalence of MetS was over 30% in recent years and increased significantly from 2011 to 2016 in young adults aged between 20 and 39 years [6,7].

The treatment of MetS revolves around controlling various risk factors, as well as lifestyle interventions [1,8]. Weight loss is one of the major methods, which often requires the regulation of total energy intake from diet and the enhancement of physical exercise simultaneously [9,10]. Even though the normalization of metabolic disorders might not have been the ultimate purpose of physical activity interventions to improve body fitness, several experimental and observational studies have shown that the improvement in fitness or increased PA helped to regulate the biomarkers of MetS to a healthier level [11,12,13,14]. In particular, a large number of research has focused on moderate-and-above-intensity PA to assess how an individual’s activity level might control the risk factors [15,16,17]. For example, the WHO 2020 guideline recommended at least 150 min/week of moderate-to-vigorous physical activity (MVPA) for health benefits [18].

Sex is an important factor when discussing metabolic-related topics. An array of research conducted worldwide has documented differences between adult males and females in the prevalence of MetS, as well as in the components of MetS including waist circumference, triglycerides, high-density lipoprotein (HDL) cholesterol, blood pressure, and fasting glucose [19,20,21,22,23,24,25]. Some of these reports displayed the association between PA and MetS by sex separately, but did not report or study the interactive effect between sex and PA on MetS. In addition, as one’s basal metabolic rate and PA levels change with age, the relationship between PA and MetS may also vary with age [26,27,28]. To address these two issues, we investigated whether and how sex and age modify the association between PA and MetS in the US population using the Nutrition Examination Survey (NHANES) data from 2003–2006 as PA was measured using an accelerometer during this period and would provide a more accurate estimation of the subjects’ activity state.

## 2. Materials and Methods

### 2.1. Study Design

The Centers for Disease Control and Prevention (CDC) designed the National Health and Nutrition Examination Survey (NHANES) to collect health- and nutrition-related information from the non-institutionalized civilian resident population of the United States (US). From 1999, the NHANES adopted a complex, multistage probability sampling design to select approximately 5000 individuals that were representative of the entire US population to participate in the study every year. The data were released in two-year cycles to the public. The NHANES consisted of three primary components: a household interview, examinations and interviews at mobile examination centers (MECs), and post-MEC data collection. For this study, demographic and lifestyle characteristics including age, sex, race/ethnicity, smoking, and drinking status were obtained at the household interview. At the onsite MEC examinations, the participants’ anthropometry (including height, weight, and waist circumference) and blood pressure were measured as components of a physical examination, and blood specimens were collected to test their triglycerides, HDL, and blood glucose. The physical activities of the selected participants were assessed using an accelerometer during the post-MEC period. In addition, a 24 h dietary recall at the MECs was conducted to estimate the daily total energy intake. More details about the NHANES protocol and study procedures are available online (https://wwwn.cdc.gov/nchs/nhanes/) (accessed on 30 December 2020).

### 2.2. Sample

This study included the NHANES 2003–2004 and 2005–2006 cycles, where the physical activity monitor (PAM) module was available. Among the 9515 individuals aged ≥20 year who participated in the MEC module, 4372 could not be clearly determined as having met the criteria of MetS, predominantly due to missing triglycerides and fasting blood glucose, and 1852 failed to meet the minimum wear time standards of the PAM. The final sample consisted of 4533 participants, after excluding another 115 participants with either extreme/implausible values (e.g., body mass index >100 kg/m^2^, daily intake <500 kcal or >6000 kcal) or who did not complete the 24 h dietary recall.

### 2.3. Accelerometer-Based Physical Activity

Physical activities were measured using an ActiGraph accelerometer (model 7164; ActiGraph, LLC, Fort Walton Beach, FL, USA). The ActiGraph counted acceleration in the vertical direction as physical activity counts and collected the records in a 1 min epoch. The participants were asked to wear the device on the hip for 7 consecutive days and to remove it only when swimming or bathing; thus, for each participant, the PAM data consisted of 1440 × 7 PA counts from the first minute of the first calendar day to the last minute of the seventh calendar day. The ActiGraph data were screened to identify non-wear periods, defined as ≥60 consecutive minutes of zero acceleration, with the allowance of up to 2 min of non-zero counts. After removing the non-wear time, subjects with at least 1 valid day, i.e., a minimum accelerometer wear time of 10 h per day, were included in this study. To classify the time spent in different activity intensities, we used cutoff methods that have been broadly accepted in previous NHANES research [12,29,30]. Specifically, MVPA was defined as ≥2020 counts per minute of activities, and for each participant, the time of MVPA was calculated as the mean of the daily total MVPA minutes across all valid days.

### 2.4. Metabolic Syndrome

The harmonized definition of MetS published in 2009 was used in this study, which incorporated medical treatments into the National Cholesterol Education Program’s Adult Treatment Panel III report (NECP/ATP III) clinical criteria [3,4]. Subjects were diagnosed as having MetS if they met three or more of the following: (1) high waist circumference: waist circumference > 102 cm in men or >88 cm in women; (2) triglycerides ≥ 150 mg/dL or treatment for elevated triglycerides; (3) HDL cholesterol < 40 mg/dL in men or <50 mg/dL in women or treatment for reduced HDL cholesterol; (4) systolic blood pressure ≥ 130 mm Hg or diastolic blood pressure ≥ 85 mm Hg or both, or treatment for hypertension; (5) fasting glucose ≥ 100 mg/dL or drug treatment for elevated glucose. The blood pressure of the participants could be measured multiple times (up to 4) at MEC, and the mean of all the available blood pressure measurements was used as the subject’s final blood pressure. The MetS status could be ascertained with a minimum of three available data of the five components (e.g., if one did not undertake an examination of triglycerides and fasting glucose but the measurements of waist circumference, HDL, and blood pressure satisfied criteria 1, 3, and 4, then the participant can still be determined to have MetS).

### 2.5. Statistical Analysis

Many participants did not contribute triglycerides and fasting blood glucose data or failed to meet the protocol requirements of the PAM, resulting in large amounts of missing MetS responses and/or PA data. Non-parametric random forest missing value imputation based on the demographic and lifestyle information, as well as self-reported physical activity habits and history of diabetes, was therefore performed to impute the missing data and account for possible selection bias [31]. The descriptive and regression analyses used the appropriate 4 year NHANES examination sample weights for the combined data to account for the complex study design and to provide nationally representative estimates. The prevalence of MetS was estimated for each weighted MVPA time tertile (1st, <10.6 min/d; 2nd, 10.6–26.7 min/d; 3rd, >26.7 min/d) and weighted age quintile (1st, 20–29 years; 2nd, 30–39 years; 3rd, 40–49 years; 4th, 50–64 years; 5th, 65–85 years) of the sampled population. Differences in the means and percentages were examined using a weighted *t* test and weighted chi-square test, respectively. Multivariable logistic regression was used to estimate the odds ratios (ORs) of having MetS per unit change in the log-transformed MVPA time. The modification effects of sex and age on the association between MVPA time and MetS were examined by including and testing the two-way and three-way interaction terms of MVPA time, sex, and age in the model after adjusting for race/ethnicity, BMI, smoking status, drinking status, and total daily energy intake. All of the analyses were conducted using R version 4.2.2.

## 3. Results

The population prevalence of MetS was 35.1%, with males being higher (37.6%) than females (32.7%). Descriptive summaries of the sample and population are shown in Table 1. The demographic and lifestyle characteristics differed significantly (all *p* < 0.001) between the populations with and without MetS. In particular, shorter mean daily MVPA times were observed in the MetS (16.5 min/d) versus the non-MetS population (26.3 min/d). According to Figure 1, among those diagnosed with MetS, the most common combination of MetS components was high blood pressure, high triglycerides, high waist circumference, and high fasting glucose. Both high waist circumference and high fasting glucose appeared in the top five most frequent combinations and were also the top two most prevalent single components. The prevalence of MetS was generally lower in women than in men across all age groups, although the sex difference varied across age groups (Figure 2). When the MVPA tertile levels were low (1st tertile) or medium (2nd tertile), the prevalence of MetS among men increased by age and reached a maximum around 40–50 yrs before decreasing. At high MVPA levels (3rd tertile), however, the prevalence of MetS among men increased from the youngest to the oldest age group. This pattern of an increasing prevalence with age was also evident among women, regardless of their MVPA level.

The changes in the prevalence of MetS for men and women by age and MVPA levels, displayed in Figure 2, suggest potential modifying effects of sex, together with age, on the association between MetS and MVPA. This was confirmed by the significant likelihood ratio test of the three-way interaction term between sex, age, and MVPA in the logistic regression model (*p* = 0.018). The predicted ORs of MetS per unit increase in the log-scaled MVPA time, shown in Figure 3, had an increasing trend from around 0.5 at age = 20 years to 1.0 at age = 80 years, indicating that the protective effect of MVPA on MetS decreased with age. Strong associations between prolonged MVPA times and a decreased odd of MetS were observed in young adults (e.g., at age = 25 years OR = 0.65, 95% CI: [0.54, 0.78] for women; OR = 0.46, 95% CI: [0.39, 0.55] for men). Moreover, the protective effect of MVPA was stronger for young men than for young women (OR = 0.73, 95% CI: [0.57, 0.93] comparing males vs. females at age 25 years in contrast to OR = 1.00, 95% CI: [0.88, 1.16] at age 60 years), but the rate of change in the ORs with age was also higher in males than in females. MVPA’s protective effect on MetS was the same for both men and women by around age = 60 years, and its effect was no longer beneficial at ages 65 years and 70 years for males and females, respectively. Figure 4 compares males versus females in their effect on MetS as a function of MVPA at the medians of the five age quintiles. This figure not only shows that males have, in general, a stronger protective effect on MetS with increasing MVPA, but more importantly, it shows that before age = 50 years, the male advantage was stronger at low MVPA levels than at high MVPA levels, as evidenced by the significant negative slopes in the first row of the panel-graphs. However, with an increase in age, the male advantage at low versus high MVPA levels attenuated. By age = 60 years, the protective effect of MetS of men over women became much smaller and did not change with the amount of MVPA. By age = 75 years, the effect on MetS in men versus women even increased with the MVPA time, although the sex difference was not statistically significant.

## 4. Discussion

The main purpose of this study was to examine whether and how sex and age modify the association between accelerometer-based PA and MetS among US adults. In general, the prevalence of MetS increased with age in both sexes, and the trend of change varied across MVPA levels. We found that males tended to benefit more than females with the same amount of MVPA, as MVPA lowered the odds of MetS in both sexes of young or middle age, and that age modified the interactive effect between MVPA time and sex. This study is novel in that the interactive effect between sex and MVPA time on MetS and how it was modified by age has not been previously examined in a large, representative sample of US adults using an objective measure of PA.

The age-specific prevalence of MetS by sex was consistent with previous research focusing on the US population [32]. Regardless of MVPA level, the prevalence of MetS in US adults aged 20–85 year increased with age, and males younger than 60 had a higher risk of MetS compared with females of the same age, and females became more vulnerable thereafter. A similar trend of sex differences in the prevalence of MetS was also reported in a recent study on the Chinese population, although the reverse point was a little earlier, at 43 years old, which might be attributed to the race and lifestyle differences between western and eastern countries resulting in a higher prevalence of MetS in the western population [33]. 

Physical activities have been indicated to be negatively related to cardiometabolic risk factors, including those comprising metabolic syndrome, by a number of studies [34,35,36]. In particular, physical activity of moderate-or-higher-intensity has played a crucial role in reducing the risk of the corresponding comorbidities, such as diabetes and hypertension [37,38,39,40]. In this study, the prevalence of MetS consistently decreased with an increase in MVPA time in both sexes and across all age groups. Nevertheless, in contrast to females, in whom the prevalence consistently increased with age in all MVPA groups, the trend only appeared in males with the highest level of MVPA, which postponed the age of the highest risk of MetS compared to the low and medium levels of MVPA. A similar conclusion was drawn from a study on the Taiwan middle-aged population, which stated that the intervention–response relationship between the frequency of physical exercise and MetS was linear in females and exponential in males. Our results of the multivariable logistic regression involving the interaction terms of MVPA time, sex, and age suggested a significantly stronger protective effect of MVPA time in young or middle-aged men than in women [41]. Females and males were found to have different mechanism of energy expenditures by previous studies. In general, the total daily energy expenditure of females was about 10% lower than men, primarily because of the difference in body composition between the sexes [42]. According to Keim et al., even with similar fatness and relative aerobic capacity, the energy expenditure per minute was higher in males. Males were also reported to consume more energy than females involved in the same intensity of PA for the same duration, which partly explains why PA was more effective in reducing the risk of MetS in males [43,44]. Preferences of PA may also a reason for the sex difference in the relationship between PA and MetS. As men were surveyed to be more likely than women to prefer outdoor activities, Chen et al. supposed that the higher vitamin D concentration and reduced systolic pressure associated with outdoor activities would result in a higher efficiency of MetS improvement in males [41,45]. The study by Oyibo et al. showed that, relative to females, more males preferred strength training, such as crunches and planks, which may evade the collection of acceleration by the accelerometer [46]. Considering that this kind of resistive activity is more likely to be combined with dynamic exercise, the MVPA time could have been underestimated among more active males in our study, further inducing the interactive effect between MVPA time and sex. 

The physiological differences between men and women are complicated in mechanism but directly related to metabolic regulation, including body weight deposition, lipid metabolism, and insulin action. In addition, there is evidence that men and women have different cardiovascular responses to dynamic exercise based on their cardiac output and vascular tone [47,48,49]. However, a significant sex difference in the association between MVPA and MetS as a comprehensive index rather than a single risk factor was observed to diminish in the older population. Although the pathogenetic mechanism of MetS remains unknown at present, scientific studies have provided evidence that MetS is causally an endocrine disease [50]. The metabolic effects of sex steroids have been described in multiple studies. Specifically, estrogen plays a protective role in the development of MetS, while testosterone inhibits the fat deposition in visceral adipose tissue [51,52]. As people age, a trans-sexual trend appears concerning the reassignment of sex hormones, with a relative increase in estrogen and a decrease in testosterone levels in men and in the opposite direction in women [53,54]. The fact that sex hormones are involved in the energy expenditure mechanism via different body compositions of men and women and that PA declines with age, it is not surprising that the sex difference in the association between MVPA time and MetS would be different across different age groups. 

Several limitations of this study should be noted. First, the cross-sectional nature of the NHANES prevents causal inference, particularly in that having MetS may conversely weaken the motivation for being active to different degrees in males and females. Second, the accelerometer was unable to detect resistance training or complete inactivity; thus, the muscle training of the subjects might be underestimated. Finally, when exploring whether the total daily energy intake influenced the association we studied, we used a single 24 h dietary recall at the MECs to retain as many subjects as possible, while the NHANES 2003–2006 actually had another post-MEC 24 h dietary recall over the phone. Although a single recall cannot reflect personal day-to-day variability, a one-day dietary record is acceptable in depicting the population-wide energy intake with validity and reliability for studies with a big enough sample size, such as the NHANES [55,56]. There are also strengths to this study. Limited by cost or feasibility, most prior epidemiological research has only collected self-reported PA information, whereas we used objectively measured PA, which is thought to provide estimates of energy expenditure with more precision than self-reported ones [57,58]. Additionally, as over half of the subjects missed the main response or risk factors in the original sample, random forest imputation was applied to reduce potential selection bias, in contrast to the majority of studies that directly excluded subjects with missing data. Finally, to the best of our knowledge, this is the first study to examine the interaction between sex and MVPA time on MetS and the modification effect of age using a nationally representative sample, which allowed the generalization of our results to US adults.

In conclusion, the prevalence of MetS in US adults decreased with age, but with different trends in males and females, respectively, at different levels of MVPA. Both sexes in the young and middle-aged populations benefitted from MVPA. A longer MVPA time was associated with a greater decrease in the risk of MetS in young men than in women, but the sex difference reduced with age and was no longer apparent in the older populations.

## Figures and Tables

**Figure 1 healthcare-11-01059-f001:**
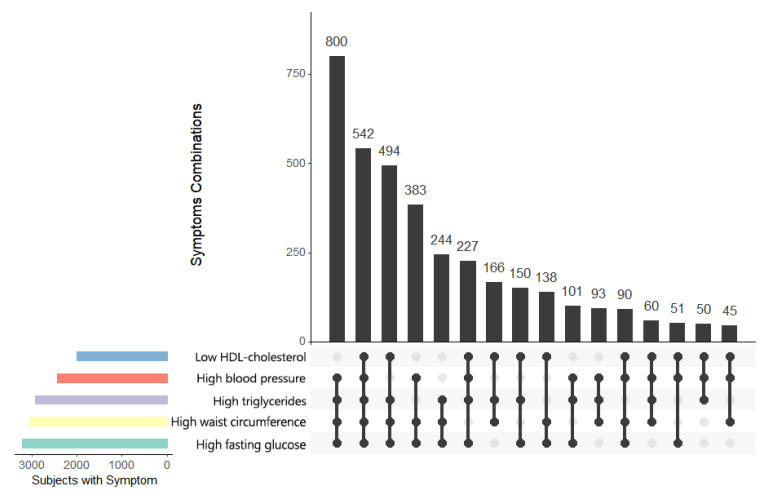
Frequencies of MetS components and their combinations among US adults ≥20 years with metabolic syndrome. Vertical bars in the main plot at top right are frequencies of subjects. Below each vertical bar is a line linking solid dots, indicating the concurrence of MetS components reflected by the bar above. Horizontal bars at bottom left indicate the frequencies of each individual MetS component.

**Figure 2 healthcare-11-01059-f002:**
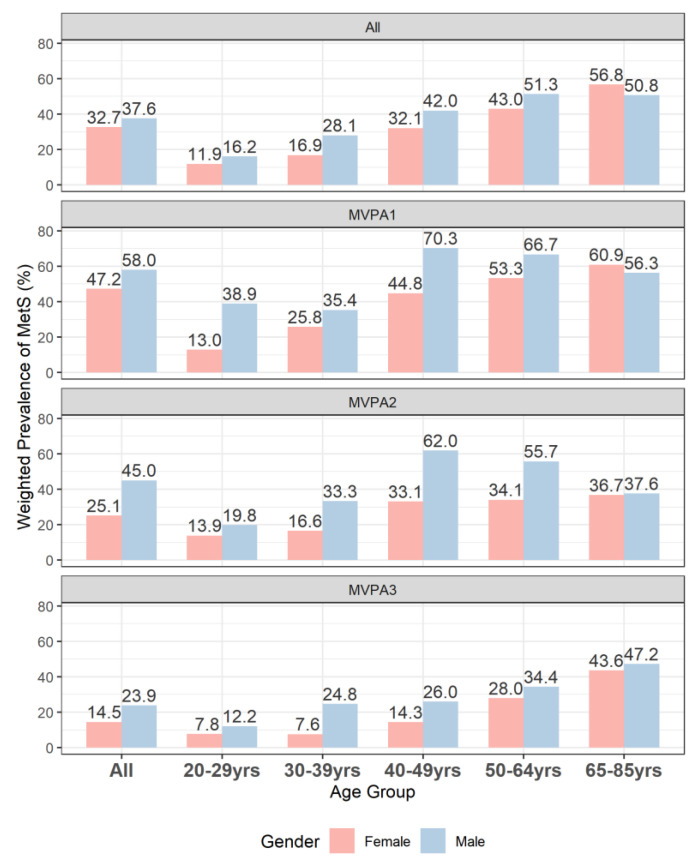
Weighted prevalence of metabolic syndrome. Bars represent metabolic syndrome prevalence (%) of male and female at each age quintile and at each tertile of MVPA time. ALL represents the case when the prevalence is not categorized by age or by MVPA time.

**Figure 3 healthcare-11-01059-f003:**
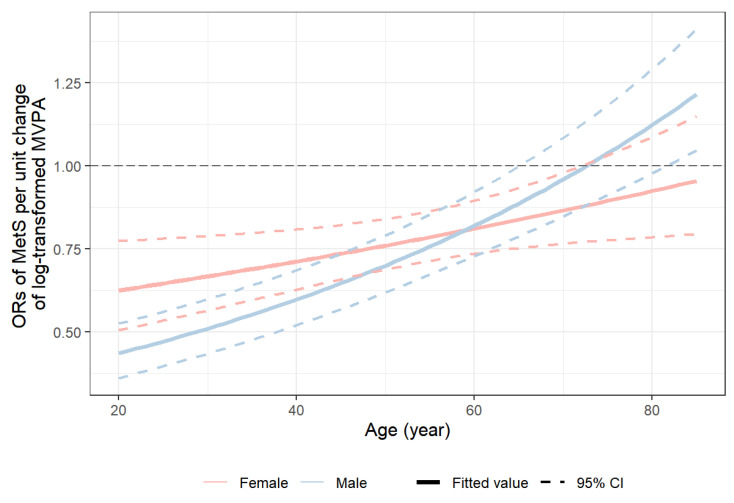
Adjusted association between MVPA time and MetS according to age. *Y* axis is the predicted odds ratio of MetS per unit change in log_e_(MVPA + 1) time in a logistic regression model with sex, age, log_e_(MVPA + 1), and their two-way and three-way interactions and adjusted for race/ethnicity (White, Mexican American, Other Hispanic, Black, and others), BMI, smoking status (never, former, and current), drinking status (non-drinker, moderate drinker, and heavy drinker), and total daily energy intake.

**Figure 4 healthcare-11-01059-f004:**
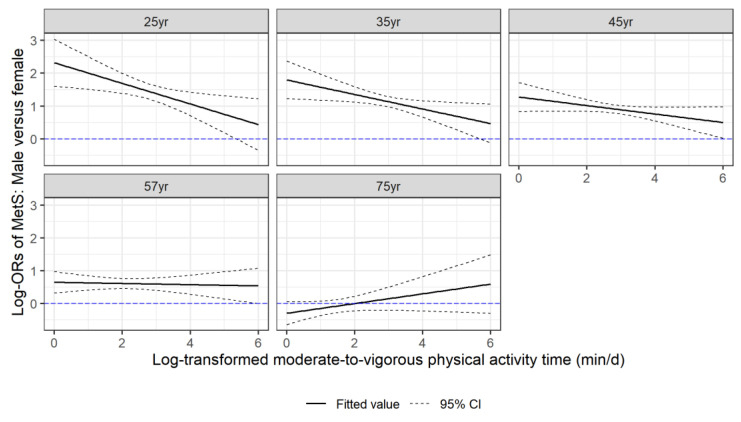
Adjusted association between sex and MetS according to log-transformed MVPA time at different age. *Y* axis is the predicted log_e_(OR) of MetS in males compared to females in a logistic regression model with sex, age, log_e_(MVPA + 1), and their two-way and three-way interactions and adjusted for race/ethnicity (White, Mexican American, Other Hispanic, Black, and others), BMI, smoking status (never, former, and current), drinking status (non-drinker, moderate drinker, and heavy drinker), and total daily energy intake. Each panel presents the relationship in the population of participants at the median age of each of the age quintiles.

**Table 1 healthcare-11-01059-t001:** Descriptive characteristics, NHANES 2003–2006.

	Total		Non-MetS		MetS	
	Sample ^1^ (*n* = 9515)	Population ^2^ (N = 207,718,631)	Sample (*n* = 6018)	Population (N = 134,893,945)	Sample (*n* = 3497)	Population (N = 72,824,686)
Age (year)	49.3 + 19.3	46.4 + 17.0	44.7 + 18.9	42.7 + 16.4	57.4 + 17.3	53.3 + 16.1
Sex						
Female	4956 (52.1)	52.0	3224 (53.6)	53.9	1732 (49.5)	48.5
Male	4559 (47.9)	48.0	2794 (46.4)	46.1	1765 (50.5)	51.5
Race/Ethnicity						
White	4894 (51.4)	71.9	3025 (50.3)	70.5	1869 (53.4)	75.1
Mexican American	1910 (20.1)	7.9	1158 (19.2)	8.1	752 (21.5)	7.3
Other Hispanic	291 (3.1)	3.5	200 (3.3)	3.7	91 (2.6)	3.1
Black	2024 (21.3)	11.4	1360 (22.6)	12.1	664 (19.0)	9.8
Other	396 (4.2)	5.4	275 (4.6)	5.7	121 (3.5)	4.7
Drinking status						
Non-Drinker	3430 (36.1)	28.9	1912 (31.8)	24.6	1518 (43.4)	36.8
Moderate Drinker	5492 (57.7)	63.4	3696 (61.4)	66.9	1796 (51.4)	56.9
Heavy Drinker	593 (6.2)	7.7	410 (6.8)	8.5	183 (5.2)	6.3
Smoking status						
Never	4909 (51.6)	50.3	3224 (53.6)	51.5	1685 (48.2)	48.0
Former	2494 (26.2)	25.0	1389 (23.1)	22.5	1105 (31.6)	29.8
Current	2112 (22.2)	24.7	1405 (23.3)	26.0	707 (20.2)	22.3
BMI (kg/m^2^)	28.6 + 6.4	28.4 + 6.4	26.6 + 5.6	26.3 + 5.4	32.0 + 6.2	32.3 + 6.3
MVPA (min/d)	20.6 + 21.5	22.8 + 21.0	24.0 + 22.7	26.3 + 21.8	14.6 + 17.6	16.4 + 17.8
Energy intake (kcal/d)	2147.0 + 861.9	2225.3 + 885.1	2223.4 + 867.3	2272.3 + 886.6	2015.5 + 836.6	2138.1 + 875.7
Waist circumference (cm)	98.3 + 15.3	97.6 + 15.7	92.7 + 13.5	91.5 + 13.3	108.0 + 13.3	108.8 + 13.6
Triglycerides (mg/dL)	149.5 + 92.1	147.7 + 92.9	116.3 + 58.3	114.0 + 58.4	206.8 + 109.7	210.0 + 111.0
HDL cholesterol (mg/dL)	55.1 + 16.2	54.5 + 15.9	59.7 + 15.7	59.0 + 15.2	47.1 + 13.9	46.2 + 13.7
Diastolic blood pressure (mm Hg)	69.0 + 13.5	70.4 + 12.6	67.3 + 12.1	68.5 + 11.1	72.1 + 15.3	74.1 + 14.4
Systolic blood pressure (mm Hg)	125.1 + 20.3	122.9 + 18.2	118.5 + 17.0	117.1 + 14.9	136.5 + 20.4	133.6 + 18.7
Fasting glucose (mg/dL)	105.1 + 28.1	102.7 + 23.9	96.7 + 19.3	95.8 + 17.2	119.6 + 34.3	115.6 + 28.9

^1^ *n* is the size of collected sample. Values in Sample column are sample mean ± SE for continuous variables and number of subjects (percentage) for categorical variables. ^2^ N is the size of population represented by the sample. Values in the Population column are sample weight adjusted population mean ± SE for continuous variables and sample weight adjusted population percentage for categorical variables.

## Data Availability

Publicly available datasets were analyzed in this study. This data can be found here: https://wwwn.cdc.gov/nchs/nhanes/continuousnhanes/default.aspx?BeginYear=2003 (accessed on 30 December 2020), https://wwwn.cdc.gov/nchs/nhanes/continuousnhanes/default.aspx?BeginYear=2005 (accessed on 30 December 2020).

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
