# Peer review of "Sex and Age Differences in Association between Physical Activity and Metabolic Syndrome: Results from NHANES 2003–2006"

_healthcare, 2023, doi:10.3390/healthcare11081059_

Round 1

Reviewer 1 Report

In this study the authors evaluated adults over the age of 20 in the United States between 2003-2006 in the National Health and Nutrition Examination Survey. They evaluated whether and how gender and age modify the association between accelerometer-based physical activity and metabolic syndrome. At baseline, this is a very important and timely study given that obesity is a growing concern in the United States and over one third of Americans are obese. The statistical analysis was well considered and performed as the authors used multiiable logistic regression was used to estimate OR of having MetS at increasing MVPA time. Modification effects of gender and age on the association between MetS and MVPA time were examined by testing for two-way and three-way interaction terms of MVPA time, gender, and age in the model after adjusting for relevant covariates.  Not surprisingly they determined that even after adjusting for demographic and lifestyle covariates, there was a significant gender difference in how increased MVPA time lowered the odds of MetS. This interactive effect also varied with age. Ultimately the authors showed that longer MVPA time was associated with more decrease in the risk of MetS in young men than women, but the gender difference reduced with age and was no longer apparent in old populations. This is a very timely and important studies. Statistically, the authors utilized a very large database and showed compelling information about the association between accelerometer-based physical activity and metabolic syndrome. Ultimately it behooves us all to improve our physical activity, men more so than women, but eventually the gender risk reduced with age. I recommend this manuscript for publication without reservation. 

Author Response

Thanks for your review!

Reviewer 2 Report

1. The no-contribution of accelerometers to resistance PA, should be controled. 

2. Waist perimeter ISN'T the only and MOST important anthropometric variable in to predict visceral fat/obesity. Waist sagital diameter and hip perimeter should be considered.

3. Fig. 1, It's a little difficult to understand...

Reviewer 3 Report

Overall, I found the manuscript very well written and interesting to read. I applaud the authors for their effort in crafting a quality manuscript. 

Please find the comments/suggestions below. These are offered in an effort to enhance the product, not judge the authors choices. 

General – I do not believe ‘gender’ is the correct term. For research purposes, ‘sex’ is the correct term. Sex is a biological fact. Gender is an identity, psychosocial or cultural factor. This is my greatest concern with the manuscript. My suggestion is to use the term 'Sex' throughout the manuscript. 

If the term Gender was used in the collection of data, a greater question arises as to the accuracy of the information collected as an individual could identify as a female, but be a biological male. Thus presenting characteristics of a male while recording data as a female. This would be a significant challenge to the accuracy of data and thus the analysis. This is not a challenge to the authors, merely a call to focus the delineation in the data. 

Abstract – 

Suggestion: delete the parenthetical numbers, keep the key words; Purpose, Methods,…

L31- change ‘old’ to ‘older’

Introduction-

L49 – change ‘proved’ to ‘indicated’ or ‘shown’

Methods – 

Well done! 

Results-

Well done! 

Discussion – 

L238 - change ‘proved’ to ‘indicated’ or ‘shown’

L269-273 – this seems a long and potentially run on sentence

When offering limitations, the ownership of the information changes from an abstract ‘this’ to a pointed ‘our’. I would suggest remaining abstract.

References – 

Appear consistent. Well done. 

Reviewer 4 Report

This is an interesting topic and has its novelty and importance. Thank you for giving me the opportunity to review this manuscript. Despite the efforts of the authors to present a clear idea about the experiment performed, I have a few concerns about this paper that I request that the authors address before the manuscript is ready for publication.

In general, the introduction is well presented, but a greater analysis of studies on the subject is lacking. Could be useful, to present additional information from previous studies using the same dataset. Another important issue is the year of the data: The data are quite old, is there any updated information to be considered? How does this issue affect the validity of the results? What changes (political, economic, organizational) could influence the validity of results from today? For the discussion, the authors need to expand, presenting some comparisons with previous data. Also, could be interesting to provide some practical applications of the results.

Round 2

Reviewer 4 Report

No additional comments.